

# The burying and grazing effects of Plateau pika on alpine grassland are small: A pilot study in a semi-arid basin on the Qinghai-Tibetan Plateau

Shuhua Yi[1*], Jianjun Chen[1, 2], Yu Qin[1], Gaowei Xu[1]

1State Key Laboratory of Cryosphere Sciences, Cold and Arid Regions Environmental and Engineering Research Institute, Chinese Academy of Sciences, 320 Donggang West Road, Lanzhou 730000, China

2 University of Chinese Academy of Sciences, No.19A Yuquan Road, Beijing 100049, China

*Correspondence to*: Shuhua Yi (yis@lzb.ac.cn)

## Abstract

There is considerable controversy about the role of Plateau pika (*Ochotona curzoniae*) in alpine grassland on the Qinghai-Tibetan Plateau (QTP). It is on one hand considered as a keystone species, on the other hand poisoned. Although significant amount of efforts have been made to study the effects of Plateau pika at a quadrat scale (~m$^2$), our knowledge about its distribution and effects at a larger scale is very limited. In this study, we investigated the direct effects, i.e. burying and grazing, of pika by upscaling field sampling at a quadrat scale to a plot scale (~1,000 m$^2$) by aerial photographing. Altogether, 168 plots were set on 4 different types of alpine grassland in a semi-arid basin on the QTP. Results showed that: 1) the effects of burying by pika piles on the reduction of vegetation cover, biomass and soil carbon/nitrogen were less than 10%, which was much smaller than the effects of bald patches; and 2) pika consumed 8-21% of annual net primary production of grassland. We concluded that the direct burying and grazing effects of pika on alpine grassland were minor in this region. Quadcopter is an efficient and economic tool for long-term repeated monitoring over large regions for further understanding the role of pika.



## 1. Introduction

Alpine grassland is important for animal husbandry and occupies about 2/3 of the total area of the Qinghai-Tibetan Plateau (QTP), but about 1/3 of this resource has degraded over the last few decades (Li et al., 2011). In addition to overgrazing (Zhang et al., 2014), climate warming and permafrost degradation (Wang et al., 2008; Yi et al., 2011), small mammals, especially Plateau pika *(Ochotona curzoniae)*, are considered an important cause of grassland degradation.

Plateau pika (hereafter pika), a small lagomorph, is believed adversely affecting alpine grassland by consuming biomass, destroying the sod layer, burying vegetation with excavated soil and expediting carbon dioxide emission (Qin et al., 2015a). The bald patches created by pika activity may increase in size over time because of erosion by wind and/or water (Wei et al., 2007). According to Shang and Long (2007), 16-54% of degraded grassland is severely degraded, the so-called "black soil patch", half of which is caused by pika (Li and Sun, 2009). For this reason, local government considers pika a pest of alpine grassland and has initiated campaigns to eradicate it since 1958 (Wilson and Smith, 2014). On the other hand, pika is believed to benefit alpine grassland by increasing infiltration, decreasing runoff (Wilson and Smith, 2014) and increasing moisture and carbon content (Li and Zhang, 2006) in the top soil (up to a depth of 10 cm). Pika is also a keystone species on the QTP (Smith and Foggin, 1999; Lai and Smith, 2003). Some authors have suggested that pika is an indicator rather than a cause of grassland degradation; pika population increases quickly only after the grassland has already been degraded (Harris, 2010; Wangdwei et al., 2013).

Although the role of pika in alpine grassland ecology is receiving more and more attention, there have been few quantitative studies at plot scale (e.g. ~1000 m$^2$, Guo et al., 2012; Wandwei et al., 2013). Typically, studies on pika effects have compared vegetation and soil characteristics and carbon fluxes at a quadrat scale (~m$^2$) among plots with different number densities of pika burrows (Guo et al., 2012; Li and Zhang, 2006; Liu et al., 2013; Wei et al., 2007; Wilson and Smith, 2014). For example, Liu et al. (2013) investigated the role of pika in alpine steppe meadows studying 8 plots with



pika burrow exit numbers varying from 0 to 76 burrow exits/100m$^2$ and found that a
higher density of pika burrow exits was associated with lower net ecosystem
exchanges, aboveground biomass and number of species. There are different levels of
heterogeneity on grassland surfaces. For example, Wei et al. (2007) classified the
grassland surface into six types: 1) mound height > 10 cm; 2) mound height between
0 and 10 cm; 3) erosion pit between 0 and 5 cm; 4) erosion pit between 5 and 10 cm;
5) erosion pit> 10 cm; and 6) undisturbed. It is critical that measurements taken at a
quadrat scale be converted to a plot scale in order to properly quantify the role of
pika. However, it is hard and inefficient to walk around ground to count the number of
burrow exits or piles of pika *in situ* on large amounts of plots (e.g. Liu et al., 2013),
not to say to quantify their area fractions in each plot. Therefore, few studies have
quantified the effects of pika on alpine grassland at plot scale.

13        Lightweight Unmanned aerial vehicles (UAVs) have developed rapidly due to

miniaturization and low cost of various sensors and embedded computers (Salami et
al., 2014). UAVs have become a popular platform at a low cost for high precision
photography recently. Photography with cm-level resolution can be achieved using
widely-used camera (Colomina and Molina, 2014). In this study, we applied a UAV
with camera to take aerial photos and aimed to: 1) test whether pika burrow exits and
piles information can be retrieved from aerial photographs at a plot scale; if so, 2)
upscale the measurements of biomass, soil carbon and nitrogen measured at quadrat
scale to plot scale and quantitatively assess the burying and grazing effects of pika.
We did not aim to investigate in this study whether pika caused degradation of
grassland or degradation of grassland caused invasion of pika; neither to investigate
the role of pika on biodiversity, although both are very important.
**2.  Methodology**
**2.1 Study area and field work**
The study area is located in the source region of the Shule River Basin on Qilian
Mountain at the northeastern edge of the QTP, China (Figure 1 a). The area has an arid
continental climate. The average annual air temperature and precipitation are about -
4.0 °C and 200-400 mm (Chang et al., in press).There are four typical types of alpine





grassland in the study area: alpine steppe (AS); alpine steppe meadow (AStM); alpine
meadow (AM); and alpine swamp meadow (ASwM) (Figure 1 b-e). The soil moisture
ranges from dry in AS grassland to wet in ASwM grassland (Qin et al., 2014).
Accordingly, the dominant species was *Stipa purpurea* in AS grassland and *Kobresia*
*pygmaea* in ASwM grassland (Table 1). We conducted field studies with field sampling
and aerial photographing. In 2012, we made seasonal measurements of grassland
vegetation cover, which is proportional to above-ground biomass (Qin et al., 2014), on
the AS, AStM and AM grasslands. The protocol of measurements can be found in Chen
et al. (2016). Vegetation cover usually peaks during the end of July and beginning of
August (Figure 2).
**2.2 Field sampling**
For each grassland type, we delineated 4 surface types: vegetation patch; new pika pile
(with loose soil and a burrow exit nearby); old pika pile; and bald patch (Figure 3 d-g).
At end of July 2014, we randomly set up 3 quadrats with iron frames measuring 50 cm
$\times$ 50 cm on each surface type in each type of grassland (Figure 3 a). For new and old
pika pile surface types (Figure 3 f and g), the iron frames were placed so as to cover
vegetation as little as possible. We took one picture of each quadrat with an ordinary
digital camera (Fujifilm (China), 1000 megapixels) held vertically at a height of ~1.4
m (Figure 3 d-g). Five soil cores were collected on each quadrat with a stainless auger
(5 cm in diameter) down to 40 cm (Figure 3 c), and bulked as one composite sample.
Three replicates on each surface type of each grassland type were sampled.
At the beginning of August 2015, we set three round plots with radius of 14 m around
sampling place in each type of grassland (Figure 3 h). Distance between plots was over
50 m. We covered all burrow exits with soil within each plot. The number of burrow
exits which were opened was counted after 72 hours. Then we put trap on each of the
opened burrow exit, and checked whether pika was caught after 48 hours. The
experiment protocol was approved by Department of Qinghai Prataculture.
**2.3 Aerial photographing**
At beginning of August 2015, we selected 14 locations, among which 4, 4, 4 and 2
locations were in AS, AStM, AM and ASwM grasslands respectively. (Figure 1). There



were 3, 2, 0, and 0 locations on the alluvial terrace; and 1, 2, 2, and 2 locations on river
terrace. All locations are generally flat with slope less than 4°. Grassland of these
locations are used for grazing during migration between settlement and mountain areas
in May-June and September-October. Pikas of these locations are not poisoned. One
location in each type of grassland was over the above-mentioned sampling plots and
quadrats (Figure 3 a). On each location, DJI drone (Phantom 3 Professional, DJI
Innovation Company, China) was auto-piloted to 12 preset way points to take photo at
a height of 20 m with camera looking vertically down using software development kits
(Yi, submitted). Altogether 168 aerial photos were taken. The Phantom 3 Professional
is a light-weight (about 1280 g including battery and propellers) four-wheel drone. It is
equipped with an autopilot system with 0.5 m vertical accuracy and 1.0 m horizontal
accuracy. It is integrated with a Sony EXMOR Sensor (maximum image size:
4000$\times$3000) and a 3-axis gimbal. Each aerial photo covers roughly 35 m $\times$26 m (Figure
3 a and b), and each pixel covers roughly 1 cm$^2$ ground area.
**2.4 Image analysis**
For those images taken on ground, we selected the part of the image within the iron
frame and retrieved green fractional vegetation cover (GFVC) using a threshold
method based on excess green index (EGI=2G-R-B; with R, G, B being red, green
and blue bands, respectively) of each pixel. More specifically, to calculate GFVC we:
1) provided an initial value of EGI threshold and compared it with each pixel; 2) if the
EGI of a pixel was greater than the threshold, the pixel was considered a vegetation
pixel and assigned a green color; otherwise it was considered a non-vegetation pixel
and assigned a yellow color; 3) compared the classified image with the original
picture. Steps1) to 3) were iterated to adjust the threshold value until the vegetation
shapes in the classified image fit those of the original picture (Figure 4). Finally, we
calculated GFVC by dividing the number of vegetation pixels into the total number of
pixels.
For pictures taken from the air (Figure 5), the new and old pika piles were marked
manually with rectangles so as to include as little intact vegetation as possible (Figure
5). We plotted the contours of the vegetation and bald patches using OpenCv Library:





1) adjusted the EGI value until its contours fit well with the shape of the vegetation and

bald patches (Figure 5), 2) calculated the area in each contour in units of pixel using

OpenCv Library; and 3) we subtracted the number of vegetation and non-vegetation

pixels of new and old pika piles from the vegetation and bald patch contours,

respectively. To exclude very small patches, we only considered the patches with area

greater than 10 cm$^2$. The area fractions of vegetation and bald patches, new and old pika

piles were then calculated by dividing the number of pixels in each surface type by the

total number of pixels (see Figure 3b).

**2.5 Laboratory analysis**

Soil samples were processed in the following steps: 1) air-dried in natural condition

avoiding direct sunshine; 2) the gravel, >2 mm in size, was sieved, separated and

weighted by electronic balance (0.01g); 3) the remaining soil samples with diameter

less than 2 mm were ground to pass through a 0.25 mm sieve and were then sent to

Lanzhou University for analysis of soil organic carbon (SOC) and total nitrogen (TN)

concentration. A detailed description of the analysis methods for SOC and TN can be

found in Qin et al. (2014).

**2.6 Data analysis**

**2.6.1 Plot scale biomass, soil organic carbon and total nitrogen**

Based on the relationship between GFVC and aboveground biomass (AGB) at

quandrat scale, established using datasets of the same study area (Qin et al., 2014), we

calculated AGB (kg/ha) =21.6×GFVC for each of surface type. For each plot, we

calculated the overall AGB with the following equation:

$$AGB_{plot} = AGB_{np}f_{np} + AGB_{op}f_{op} + AGB_{bp}f_{bp} + AGB_{vp}f_{vp} \quad (1)$$

Where plot, np, op, bp, and vp represent plot, new pika pile, old pika pile, bald and

vegetation patches, respectively; f represents area fraction (%) of each surface type.

The SOC and TN at plot scale were then calculated in a similar way as that of AGB.

We defined the effect of each surface type ($E_{type}$) on AGB reduction of grassland as:

$$E_{type,agb} = \frac{(AGB_{type} - AGB_{vp})f_{type}}{\sum[(AGB_{type} - AGB_{vp})f_{type}]} \times 100 \quad (2)$$

Where $f_{type}$ represents the area fraction of a surface type in a plot (%), $\sum$ means the



sum. For the vegetation patch surface type, $E_{type}$ equals 0 and has no effect in AGB
reduction. The higher the value of $E_{type,agb}$, the higher the effect of a surface type on
plot-scale AGB reduction. The effects on SOC and TN reduction were calculated in a
similar way. The burying effects from pika piles were calculated as the sum of $E_{np}$ and
$E_{op}$.
**2.6.1 Plot scale pika number and grazing effects**
Two ratios were used in calculating number of pika from aerial photos at plot scale.
First was the ratio (r1) between the number of in-use burrow exits and the total number
of burrow exits, and the ratio (r2) between the number of pikas caught and the number
of in-use burrow exits, both of which were developed using field data for each grassland
type (Figure 3 h). We then calculated the number of pika in a plot covered by each aerial
photo (Figure 3 b) with these two ratios and the total number of pika piles delineated
from each aerial photo (Figure 5; equation 3).
$N_{pika} = N_{pile} \times r1 \times r2$ (3)
Where $N_{pika}$ and $N_{pile}$ are the number of pika and the number of total pika piles in
a hectare, respectively.
Each pika consumes ~8.06 kg of grass dry matter per year (Hou, 1995; equation 4).
Pika affects above-ground biomass more than root system (Sun et al., 2016). The annual
primary production of grassland roughly equals to peak time aboveground biomass
($AGB_{plot}$; Scurlock et al., 2002). Finally, we estimated the effects of direct graze
consumption by pika ($E_{graze}$, %) in a plot (Equation 5).
$AGB_{pika} = N_{pika} \times 8.06$ (4)
$E_{graze} = \dfrac{AGB_{pika}}{AGB_{plot}} \times 100$ (5)
$AGB_{pika}$ is the biomass consumed by pika (kg/ha).
The data were presented as mean ± standard deviation. Statistical analyses were
performed using the SPSS 17.0 statistical software package (SPSS Inc., Chicago, IL,
USA). One-way analysis of variance (ANOVA) and a multi-comparison of a least
significant difference (LSD) test were used to distinguish between differences at the
p=0.05 level.



## 3. Results

### 3.1 Quadrat scale characteristics

The GFVCs of the vegetation patches were greater than 60% for both AM and ASwM grasslands, while those of AS and AStM grasslands were less than 30% (Figure 6a). The GFVC of vegetation patches was significantly greater than that of other surface types for most of the grasslands (p<0.05). Because some vegetation was included in the $50 \times 50$ cm iron frame, the GFVC of new pika pile was not zero, but was usually less than 10%. Vegetation also grew on the piles, so the GFVC of old pika pile was usually greater than that of new pika pile. Bald patch GFVC was similar to that of new pika pile.

The SOC/TN densities of 40 cm soil column ranged between 3.5/0.45 and 8.0/1.2 kg/m$^2$ (Figure 6b and c). Both SOC and TN densities under vegetation patches were significantly greater than those under bald patch (p<0.05). SOCs under vegetation patches of 3 out of 4 grasslands were significantly greater than those under new and old pika piles (Figure 6b). TNs under vegetation patches were only significantly greater than those of new and old pika piles on the ASwM grassland (Figure 6c). Species in vegetation patches were dominant by palatable species, while forbs with low-nutrient were common on bald patches and old pika piles on all 4 different grasslands (Table 1).

### 3.2 Area fractions and numbers of surface types at plot scale

Except for the ASwM grassland, the mean area fractions of vegetation patches were about 30%, and significantly less than bald patches (p<0.05; Figure 7a). The mean area fractions of new and old pika piles were all less than 2% for all grasslands (Figure 7b). The mean number of patches of vegetation (bald) patches ranged from ~33,000/ha (17,000/ha) in AM grassland to ~100,000/ha (67,000/ha) in AStM grassland (Figure 7c). The mean number of new (old) pika piles ranged from ~130/ha (160/ha) to ~270/ha (400/ha, Figure 7d).

### 3.3 Effects of surface types at plot scale

Due to the large area fractions of bald patches (Figure 7a) and low vegetation cover (Figure 6a), the effects of bald patches on reduction of above-ground biomass ranged from 80% on ASwM grassland to 98% on AS and AStM grasslands (Figure 8a). The



effects of pika piles were significantly less than that of bald patches. The soil organic
carbon and total nitrogen had the similar pattern as that of above-ground biomass
(Figure 8 b and c).
**3.4 Grazing effects of pika at plot scale**
The mean ratio between in-use burrow exits and total burrow exits (r1) ranged from
0.22 to 0.42, and there were no significant differences among different grassland types
($p>=0.05$; Figure 9a). The mean ratio between number of pikas and in-use burrow exits
(r2) ranged from 0.18 on ASwM grassland to 0.4 on AM grassland (Figure 9b). The r2
ratio of ASwM grassland was significantly less than those of the other grasslands
($p<0.05$). The mean number of pikas ranged from 27 ha$^{-1}$ to 60 ha$^{-1}$, and there were no
significant differences among different types of grasslands ($p>=0.05$; Figure 9c). The
graze effects of pika on aboveground biomass ranged from 8% to 21%, with that on
AStM significantly greater than those of the other grasslands ($p<0.05$; Figure 9d).
**4. Discussion**
**4.1 Burying and grazing effects of pika on grassland**
Previous studies indicated that pika adversely affect alpine grassland directly through
1) burying of vegetation with soil while burrowing and 2) consumption of vegetation
in competition with domestic animals for food (Yang and Jiang, 2002). However, our
study showed that both new and old pika piles accounted for only a very small area
fraction (<2%) of the total plot area (Figure 7b), showing that burying has minimal
effects on aboveground biomass, soil carbon and total nitrogen (Figure 8). The
aboveground biomass at peak growing season is usually used as surrogate of annual
net primary production (Scurlock et al., 2002). Pika only accounted for 21% at
maximum on different types of grassland on two different geomorphology (Figure
9d).
Sun et al. (2016) classified study sites into four classes, i.e. approximately zero pika
density (0-15 ha$^{-1}$), low pika density (15-110 ha$^{-1}$), medium pika density (110-200 ha$^{-1}$
), and high pika density (200-300 ha$^{-1}$). Our plots belong to the first two classes
(Figure 9 c). Due to different precipitation and temperature conditions, net primary
production, soil carbon and nitrogen exhibits strong spatial heterogeneity (Luo et al.,



2004). Therefore, to properly evaluate the direct burying and grazing effects of pika
on the QTP, large amounts of plots under different combined conditions of climate
and pika densities should be investigated.
**4.2 Effects of pika on bald patches**
There were bald patches of various sizes on the grasslands (see Figure 5), which played
a much more important role than pika piles in reducing vegetation cover, aboveground
biomass and soil carbon and nitrogen at the plot scale (Figure 8). We retrieved gravel
contours using the threshold of R+G+B and determined whether each was in a
vegetation or bald patch contour. The number of gravel contours in bald patches was
significantly greater than the number in vegetation patch contours (e.g. Figure 3 e and
5). For example, there was ~80/5 gravel/m$^2$ in bald/vegetation patches on the AM
grassland (Figure not shown). High amounts of gravel content are not beneficial for
nutrient retention and vegetation growth (Qin et al., 2015b): once the fine soil has been
eroded, vegetation in a bald patch is slow to recover (Gao et al., 2011).
Wei et al. (2007) suggested that a bald patch developed from a new pika pile through
its succession to an old pika pile and further erosion by wind and/or water. Other studies
have suggested that a bald patch originates from the collapse of a burrowing tunnel,
repeated freeze and thaw processes, trampling during grazing or some combination of
these factors (Zhou et al., 2003; Cao et al., 2010). However, none of these suggestions
have been supported by field observations (Wilson and Smith, 2014). It is, therefore,
critical to perform long-term repeated monitoring studies to determine: 1) whether bald
patches are developed from pika piles or burrow tunnels?; 2) how quickly does a bald
patch expand?; and 3) what are the major factors affecting bald patch expansion?
**4.3 Cons and pros of quadcopter in studying pika's effects**
Pika piles or burrow exits and bald patches are too numerous to be quantified easily on
ground by human; they are also too small to be identified by regularly available satellite
remote sensing data (Figure 5 and 7). Quadcopter integrated with a camera has the
following advantages in studying pika's effects: 1) large coverage. It can easily cover
an area of ~1000 m$^2$ when it is flied at a height of ~20 m, therefore, aerial photos can
be used to better characterize patches of different sizes than photos taken on ground; 2)





high resolution. Each pixel represents area if ~1 cm$^2$ when photo is taken at a height of
~20 m, which is good enough for identifying pika piles and bald patches (Figure 5); 3)
high locating accuracy. The distance between the center of an aerial photo and the
corresponding preset way point is ~1 m, which makes it feasible for repeated
monitoring over the same plots (Yi, submitted); 4) low cost. Each Phantom 3
quadcopter costs about 1,000 USD; and 5) high efficiency. In our study, it took only 2
minutes to fly to 12 preset way points and take photos automatically (Figure 3a).
Chen et al. (2016) found that the fractional vegetation cover derived from aerial photos
had better correlations with satellite normalized difference vegetation index, which is
usually used to estimate vegetation biomass (e.g. Gao et al., 2013), than quadrat-scale
photo taken on ground on patchy grassland. It is a non-destructive method to estimate
biomass or soil carbon/nitrogen at plot scale with only few samples at quadrat scale
sampled. Therefore, it is feasible to deploy quadcopter to monitor large amounts of
plots in alpine grassland on the QTP repeatedly over a long-term range.
However, we do acknowledge that there are some shortcomings of quadcopter: 1) we
cannot assess role of pika at species level with quadcopter. For example, selective
grazing behavior of pika can sometimes improve alpine grassland biodiversity (Harris
et al. , 2016 and Zhang et al., 2016), which cannot be upscaled to a plot scale in aerial
photos; 2) Quadcopter with a common camera cannot provide soil moisture information,
while the burrowing activity of pika can improve infiltration and increase soil water
content (Wilson and Smith, 2014). Therefore, both aerial surveying with quadcopter
and ground sampling should be used together to investigate the role of pika
comprehensively.
**5. Conclusions**
We up-scaled the quadrat-scale measurements of vegetation cover, biomass, soil carbon
and nitrogen of 4 different surface types, i.e. vegetation and bald patches, new and old
pika piles, to plot-scale using aerial photography. We then assessed the direct burying
and grazing effects of pika. We concluded that both the direct effects were minor on
different types of grasslands on two different geomorphology. Bald patches had great
impact on the reduction of biomass, soil carbon and nitrogen, but cannot be directly





associated with pika activity at the current stage, which requires long-term repeated
monitoring the changes of piles and burrow tunnels created by pika. Our study
suggested that it is feasible and efficient to use quad-copter to monitor large amounts
of patchy grassland plots and study the roles of pika.
**Acknowledgements**
We would like to thank Dr. Matthias Peichl of Swedish University of Agricultural
Sciences for his helpful suggestions and comments; and Mr. Xiaofeng Deng and
Tianfeng Wei on helping identifying the geomorphology of our study locations using
Google Earth. This study was jointly supported through grants from the Chinese
National Natural Science Foundation Commission (41271089, 41501081 and
41422102), and the independent grants from the State Key Laboratory of Cryosphere
Sciences (SKLCS-ZZ-2015). No conflict of interest was involved in this study.

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

Chinese with English abstract).



**Table 1.** The latitude, longitude, elevation of four different types of alpine grassland

and the dominant species on different surface types of each grassland.

| Grassland Type | Latitude, Longitude, Elevation | Vegetation patch | Bald patch | Old pika pile |
|---|---|---|---|---|
| Alpine steppe (AS) | 38°38′05.4″ 98°06′41.7″ 3768 m | *Stipa purpurea, Artemisia minor* | *Heteropappus hispidus (Thunb.) Less., Saussurea arenaria Maxim.* | *Potentilla bifurca Linn., Saussurea arenaria Maxim.* |
| Alpine steppe meadow (AStM) | 38°28′34.6″ 98°19′22.8″ 3886 m | *Carex moorcroftii, Stipa purpurea* | *Ajania tenuifolia, Potentilla bifurca Linn.* | *Potentilla bifurca Linn., Saussurea arenaria Maxim* |
| Alpine meadow (AM) | 38°25′15.2″ 98°18′30.4″ 3897 m | *Kobresia capillifolia, Carex moorcroftii* | *Glaux maritima Linn., Polygonum sibiricum Laxm.* | *Aster tataricus L. f., Polygonum sibiricum Laxm.* |
| Alpine swamp meadow (ASwM) | 38°19′56.2″ 98°13′35.1″ 4043 m | *Kobresia pygmaea, Kobresia humilis* | *Carex atrofusca Schkuh., Glaux maritima Linn.* | *Polygonum sibiricum Laxm., Veronica didyma Tenore.* |



**Figure Legends**
**Figure 1.** a) Source region of Shule River Basin and its location in the Qinghai
Tibetan Plateau; The rectangles indicate the locations of auto-piloted flight (each with
12 way points), 1-4 indicate the location of field sampling on each type of grassland;
b)-e) show aerial photographs of 4 types of alpine grasslands (AS: alpine steppe;
AStM: alpine steppe meadow; AM: alpine meadow; and ASwM: alpine swamp
meadow) investigated in this study. Each photograph covers ~ 35 m×26 m ground
area.
**Figure 2.** Seasonal variations of fractional vegetation cover over May 19-August 30,
2012 on alpine steppe, alpine steppe meadow and alpine meadow grasslands of Shule
River Basin.
**Figure 3.** a) Diagram of ground sampling and aerial photographing; b) aerial
photograph on one of 12 way points (solid black rectangles in a), each photo covers
~35 m by 26 m ground area, and was analyzed to have 4 parts, i.e. VP (vegetation
patch), BP (bald patch), NP (new pika pile) and OP (old pika pile); c) ground
sampling quadrat with 50 cm by 50 cm for vegetation cover, soil carbon and nitrogen
(open rectangles in a) with red for vegetation patch (d), black for bald patch (e), green
for new pika pile (f), and blue for old pika pile (g)); and h) a circular plot with radius
of 14 m for counting pika piles and pikas.
**Figure 4.** A photo taken on ground (left) and three examples (white rectangles) of
green vegetation (green) classification (1-3 on the right).
**Figure 5.** An aerial photo and contours of vegetation patch (red curves, VP), bald
patch (yellow curves, BP), new pika pile (red rectangles, NP), old pika pile (black
rectangles, OP) and enlarged examples on the right for each type. Pink contour
indicates gravel.
**Figure 6.** Green fractional vegetation cover (GFVC; %; a) soil organic carbon density
(SOC; $kg/m^2$; b) and total soil nitrogen density (TN; $kg/m^2$; c) of vegetation patch
(VP), new pika pile (NP), old pika pile (OP) and bald patch (BP) at a quadrat scale of
four types of alpine grasslands (see Figure 1). Error bar indicates ± standard deviation,





different letters above error bar indicate significant differences among surface types
(p<0.05).
**Figure 7.** Area fraction (%) and number (ha$^{-1}$) of vegetation patch (VP), new pika pile
(NP), old pika pile (OP) and bald patch (BP) at a plot scale of four types of alpine
grasslands (see Figure 1). Error bar indicates±standard deviation, different letters
above error bar indicate significant differences between VP and BP or between NP
and OP (p<0.05).
**Figure 8.** Effects of new pika pile (NP), old pika pile (OP) and bald patch (BP) on
reduction of fractional vegetation cover a), soil carbon density (SOC); b) and total
nitrogen (TN); c) on four types of alpine grasslands (see Figure 1). Error bar indicates
±standard deviation, different letters above error bar indicate significant differences
among different surface types (p<0.05) .
**Figure 9.** a) ratio between in-use burrow exits and total burrow exits (r1); b) ratio
between number of pika and in-use burrow exits (r2); c) number of pikas (ha$^{-1}$); and
d) effects of pika grazing on above ground biomass (%) on four types of alpine
grasslands (see Figure 1). Error bar indicates±standard deviation, different letters
above error bar indicate significant differences among different grassland types
(p<0.05) .





1   **Figure 1.**

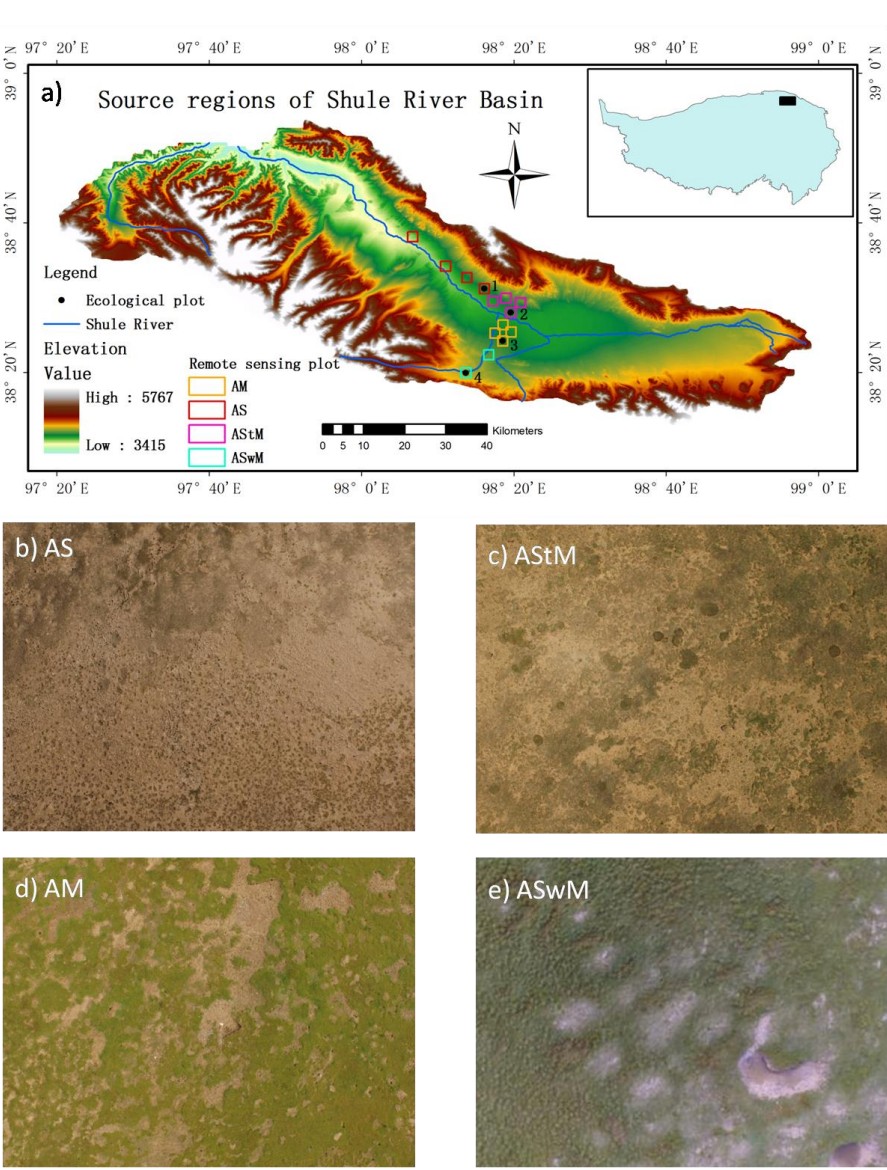





1    **Figure 2.**

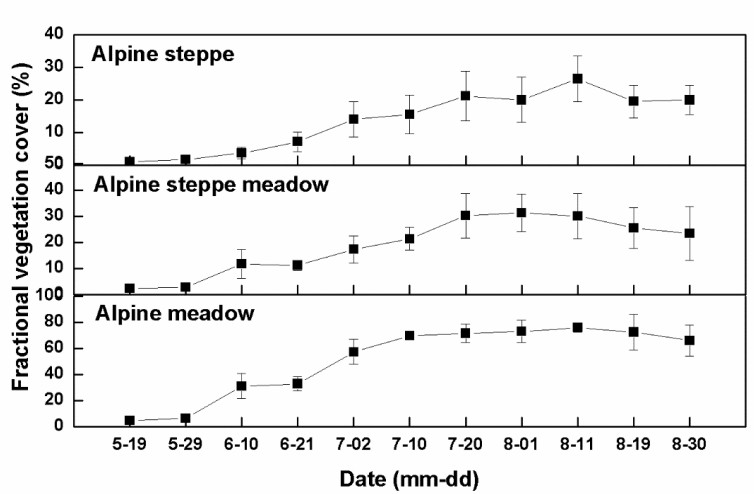



**Figure 3.**

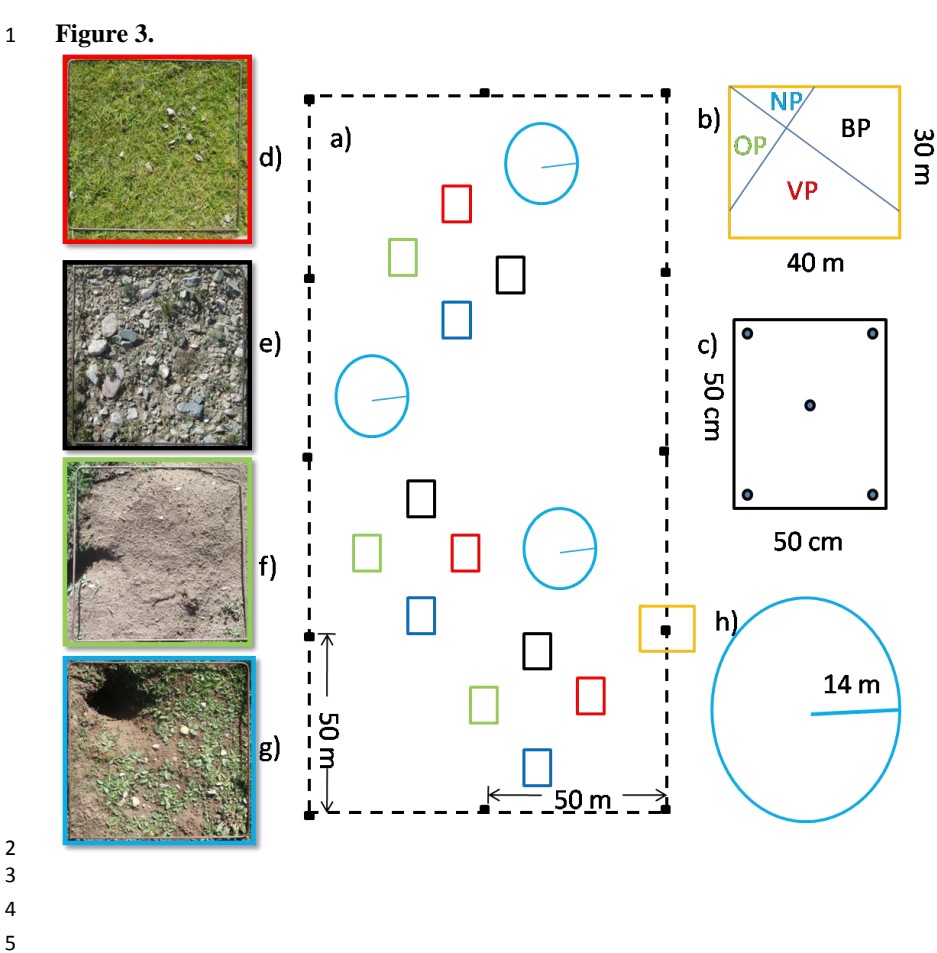




1   **Figure 4.**

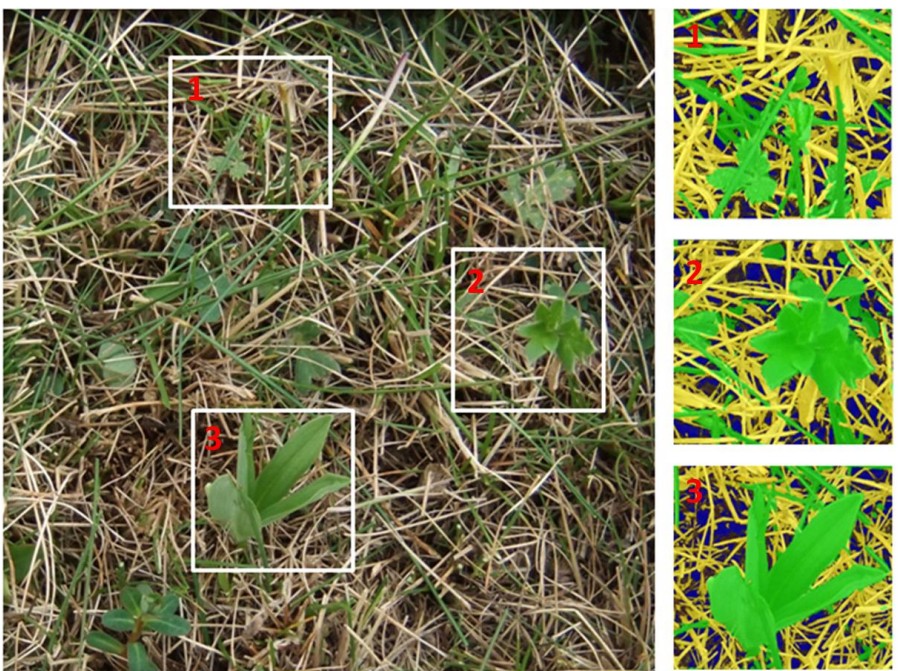





1    **Figure 5.**

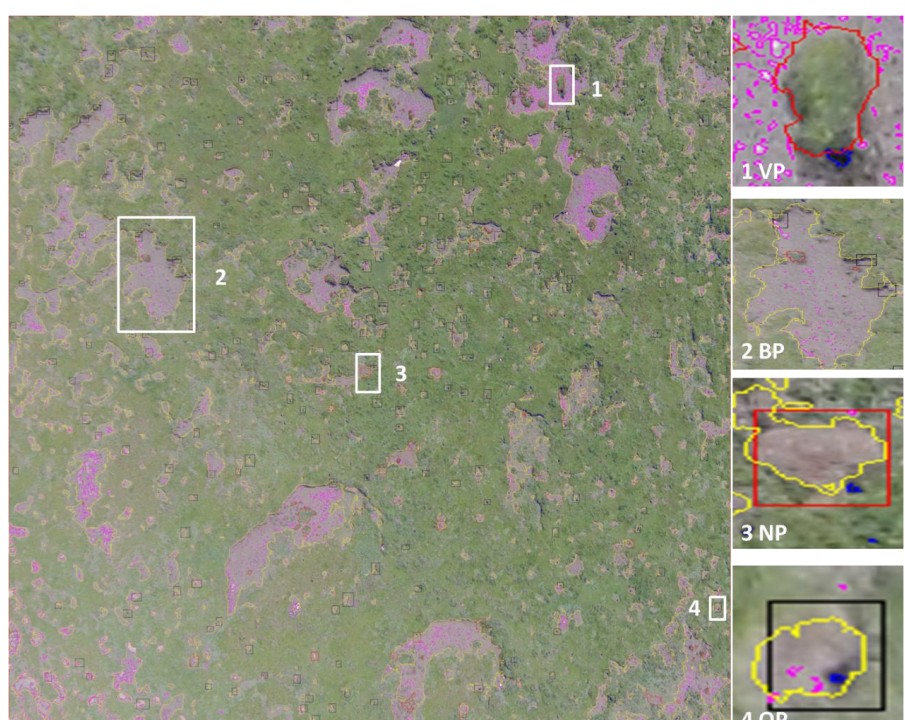

3



1 **Figure 6.**

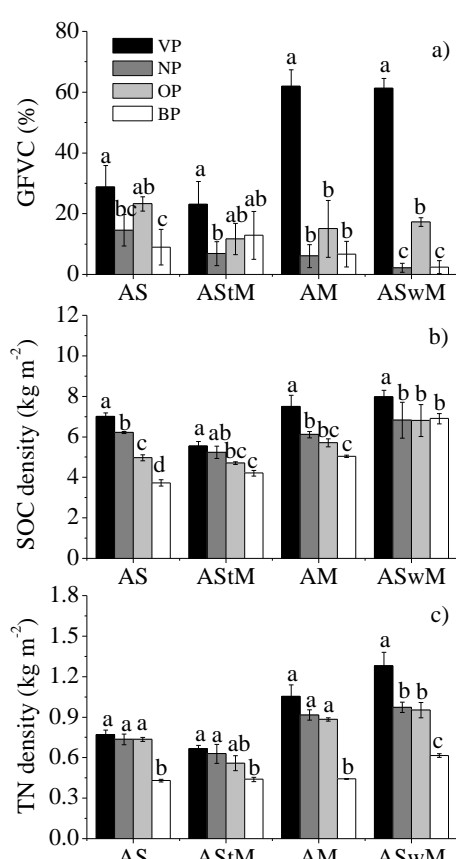



1 **Figure 7.**

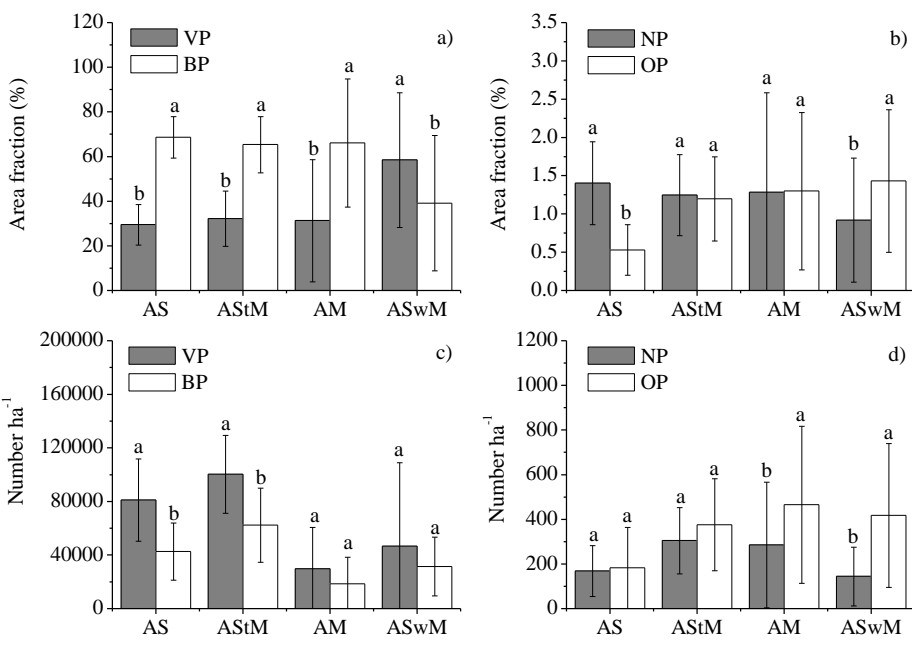



**Figure 8.**

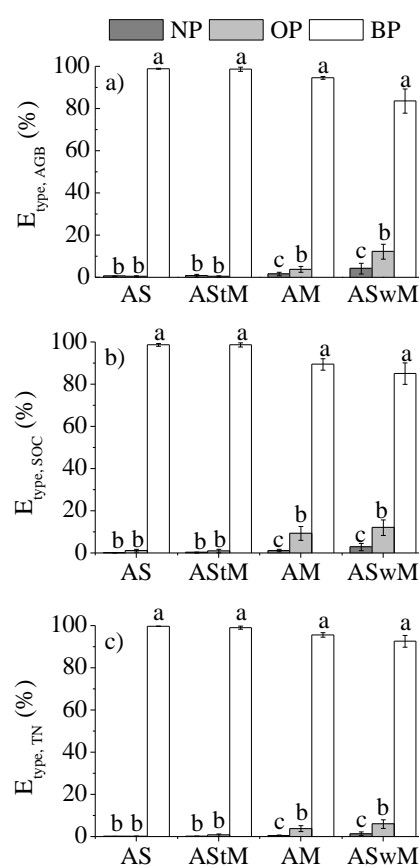






1 **Figure 9.**

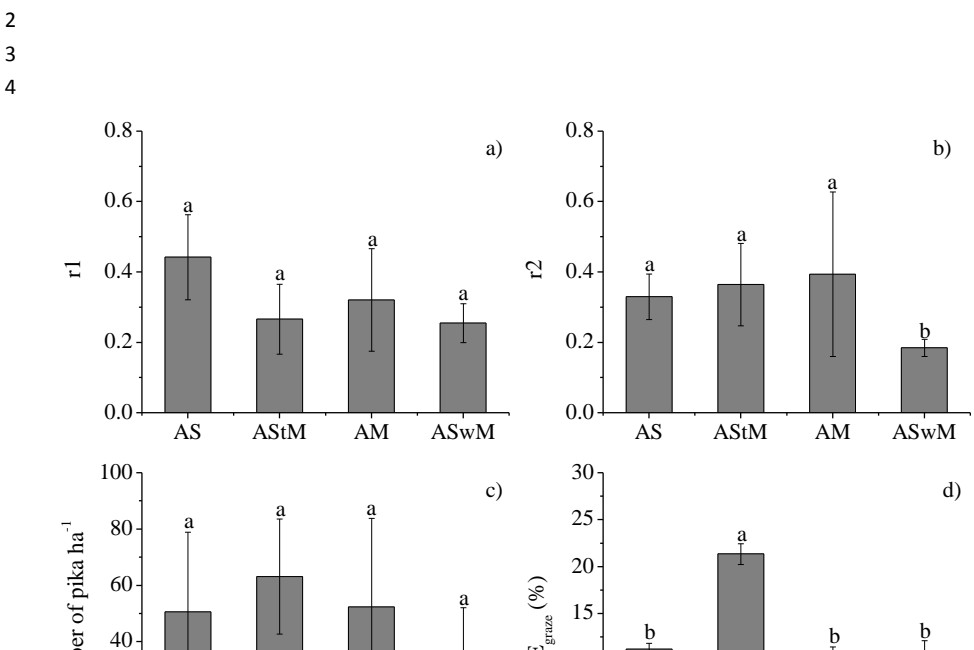

