# Peer review of "The burying and grazing effects of Plateau pika on alpine"

_Biogeosciences, 2016_

## Referee Comment (RC1) · Anonymous Referee #1 · 8 Jul 2016

This article demonstrates the advantage of UAV for monitoring a keystone species "pika" on alpine grassland of Qinghai-Tibetan Plateau. The article is well organized and interesting results are demonstrated, thus this article is suitable for publish in Biogeosciences. However, particularly in data analyses part in UAV captured images, they seem like too subjective and too classical, in a point of view remote sensing data processing technique. Thus the reviewer feels to need additional data analyses for the improvement of interpretation of the beheivor of "pika" on alpine grassland. Therefore, the reviewer 's recommendation is major revision for this article.

1.Need more objective analyses on UAV's image data processing The authors processed UAV's fine-spatial resolution's data as: identify green fraction areas, (based on

green fraction) AGB, SOC, TN estimate by the empirical relation of laboratory analysis. To estimate green fraction areas are based on try & error threshold method by the data of R, G, B signals. The reviewer suggests to vegetation indices (VIs) estimation by bands data used formula, such as NDVI type. As the author's know, most of VIs is consist of combination of red and near-infrared (NIR) signal, but GRVI (green – red)/(green + red) is also one of good indicator to monitor vegetated area's status. For example, if GRVI is higher, target pixel has (maybe) multi-layer structure of grass. By assist of GRVI type VIs, maybe reduce the uncertainty for the conversion from green area fraction (this information only have "one" layer information, i.e., yes or no) to AGB. In addition, supervised and/or unsupervised classifications are also obtained another useful information about the statistical base (e.g., size distribution of patches in which same classified). Maybe two additional data analyses (GRVI type VIs estimation and apply classification for UAV's images) indicate new-insight.

Another comments P4 L18 Fujifilim (China) –> Fujifilm (Japan) P4 L27 experimental protocol was approved by Department of Qinghai Prataculture –> If can cited document are available (as written in Chinese), please cite in reference.

4.1. Burying ... It is better to change order paragraph stared from P9L26- (Sun et al. (201&) ..) to easy to read.

---

## Referee Comment (RC2) · Anonymous Referee #2 · 15 Jul 2016

Review of "The burying and grazing effects of Plateau pika on alpine grassland are small: A pilot study in a semi-arid basin on the Qinghai-Tibetan Plateau" by Shuhua Yi and others

In this manuscript, the authors are trying to show the advantages of the UAV remote sensing for evaluation of plant biomass. Presented data and descriptions include important information. The aim and concept of the study presented in this manuscript is interesting. However, there are several weaknesses before considering a publication as an original article in Biogeosciences. I hope following comments will help the authors to improve the contents and discussions.

1. Relationship between title and main contents of the text: As the authors themselves

said in the final sentence of the introductory section, their aim is not to describe the effects of pika activity on grassland degradation. But, the title is telling the conclusion of this question. This kind discrepancy was fatal for the article. In general, "abstract" is describing the essence of the main text, and "title" is expressing the essence of the abstract, meaning the "conclusion". The aim of this manuscript in accordance to the line 18-21 is to show the effectiveness of the aerial photos taken by the UAV for investigation of vegetative conditions. The title should be changed to fit the main point of this manuscript. Also the parts of the abstract and concluding section should be revised to match the aim and conclusion.

2. Methodology of the image processing for identification of green fractions of plots: The authors used the aerial photo data of R, G and B signals for estimating green fraction area of each plot using fairly conventional technique based n trial and error threshold. At least, the authors had to consider to use vegetation indices that have been commonly (almost universally) used, such as NDVI.

3. Individual points: P3, L13-15: This sentence was unreasonable. Rapid development of lightweight UAVs has been made by multiple demands. Usage for remote sensing is just one of those demands. P3, L18-21: If the test in the aim 1) could not provide the favorable results, the authors could not execute the analysis in aim 2). In this case, they could not reach the conclusion described in the title and abstract. Namely, "if so" in between 1) and 2) is unnecessary.

P4, L9-10 and Figure 2: The method or techniques for obtaining the values of vegetation cover should be explained in the text.

P7, L6: "2.6.1" => "2.6.2"
* * *

---

## Author Comment (AC1) · 2 Sep 2016

**Reviewer 1.**

This article demonstrates the advantage of UAV for monitoring a keystone species "pika" on alpine grassland of Qinghai-Tibetan Plateau. The article is well organized and interesting results are demonstrated, thus this article is suitable for publish in Bio-geosciences. However, particularly in data analyses part in UAV captured images, they seem like too subjective and too classical, in a point of view remote sensing data processing technique. Thus the reviewer feels to need additional data analyses for the improvement of interpretation of the beheivor of "pika" on alpine grassland. Therefore, the reviewer 's recommendation is major revision for this article.

1.Need more objective analyses on UAV's image data processing The authors processed UAV's fine-spatial resolution's data as: identify green fraction areas, (based on green fraction) AGB, SOC, TN estimate by the empirical relation of laboratory analysis. To estimate green fraction areas are based on try & error threshold method by the data of R, G, B signals. The reviewer suggests to vegetation indices (VIs) estimation by bands data used formula, such as NDVI type. As the author's know, most of VIs is consist of combination of red and near-infrared (NIR) signal, but GRVI (green –red)/(green + red) is also one of good indicator to monitor vegetated area's status. For example, if GRVI is higher, target pixel has (maybe) multi-layer structure of grass. By assist of GRVI type VIs, maybe reduce the uncertainty for the conversion from green area fraction (this information only have "one" layer information, i.e., yes or no) to AGB. In addition, supervised and/or unsupervised classifications are also obtained another useful information about the statistical base (e.g., size distribution of patches in which same classified). Maybe two additional data analyses (GRVI type VIs estimation and apply classification for UAV's images) indicate new-insight.

Reply:

**Thank you for your suggestion. We used both the green relative vegetation index (GRVI, Figure 1) and excess green index (EGI, Figure 2) to estimate fractional vegetation cover (FVC). Although there are some subtle differences between two methods (see black rectangles in Figure 1 and 2), the overall FVCs are almost the same. Since there are small differences between the FVC values derived from both methods, we do not further establish the relationship between FVC and aboveground biomass.**

**We agree that our try & error threshold method introduced some uncertainties into the estimation of FVC. Since there is no "true" FVC values of grassland, results from supervised classification have been usually used as "true" FVC values. For example, supervised classification using WinCAM software was used in Yi et al. (2011), Ren et al. (2014) and Zhou et al. (2015). Yi et al. (2011) found that widely used visual estimation had large variations among different estimators, while WinCAM classification was time-consuming. Ren et al. (2014) found that threshold-based estimation was comparable to that of WinCAM classification and was more efficient than WinCAM.**

**Therefore, in our revised manuscript, we will not recalculate and compare FVCs using different methods, since it is not the focus of this study. However, we will discuss the threshold method using existing studies and the figures presented in this reply.**

[Figure]

Figure 1 Fractional vegetation cover estimation using green relative vegetation index (GRVI)

[Figure]

Figure 2. Fractional vegetation cover estimation using excess green index (EGI)

Another comments

P4 L18 Fujifilim (China) –> Fujifilm (Japan)

**Thank you for your suggestion, we will make the change in the revised manuscript.**

P4 L27 experimental protocol was approved by Department of Qinghai Prataculture –> If can cited document are available (as written in Chinese), please cite in reference.

**Thank you for your suggestion. The Department of Qinghai Prataculture provided an oral approval of our study, which requires very limited amount of pika-trapping work. Therefore, no formal document is available. We will explain it in the revised manuscript.**

4.1. Burying ... It is better to change order paragraph stared from P9L26- (Sun et al. (201&) ..) to easy to read.

**Thank you for your suggestion. We will change the sentences from**

**"Sun et al. (2016) classified study sites into four classes, i.e. approximately zero pika density (0-15 ha$^{-1}$), low pika density (15-110 ha$^{-1}$), medium pika density (110-200 ha$^{-1}$), and high pika density (200-300 ha$^{-1}$)."**

**to**

**"Sun et al. (2016) classified study sites into four classes: 1) approximately zero pika density (0-15 ha$^{-1}$); 2) low pika density (15-110 ha$^{-1}$); 3) medium pika density (110-200 ha$^{-1}$), and 4) high pika density (200-300 ha$^{-1}$)."**

**Reviewer 2**

Review of "The burying and grazing effects of Plateau pika on alpine grassland are small: A pilot study in a semi-arid basin on the Qinghai-Tibetan Plateau" by Shuhua Yi and others
In this manuscript, the authors are trying to show the advantages of the UAV remote sensing for evaluation of plant biomass. Presented data and descriptions include important information. The aim and concept of the study presented in this manuscript is interesting. However, there are several weaknesses before considering a publication as an original article in Biogeosciences. I hope following comments will help the authors to improve the contents and discussions.

1. Relationship between title and main contents of the text: As the authors themselves said in the final sentence of the introductory section, their aim is not to describe the effects of pika activity on grassland degradation. But, the title is telling the conclusion of this question. This kind discrepancy was fatal for the article. In general, "abstract" is describing the essence of the main text, and "title" is expressing the essence of the abstract, meaning the "conclusion". The aim of this manuscript in accordance to the line 18-21 is to show the effectiveness of the aerial photos taken by the UAV for investigation of vegetative conditions. The title should be changed to fit the main point of this manuscript. Also the parts of the abstract and concluding section should be revised to match the aim and conclusion.

**Thank you for your suggestion. There is an important research question, i.e. whether plateau pika causes degradation of grassland, or plateau pika invades after grassland degradation caused by other factors, e.g. climate warming or overgrazing. Meanwhile, changes of grassland include not only changes of biomass and soil carbon and nitrogen, but also changes of species.**

**The current study specifically quantified the burying and grazing effects of pika on grassland at a scale of ~900 m$^2$. This study cannot quantify pika's effects on species and cannot answer the cause-effect relationship between pika and grassland degradation, which requires long-term experiment studies.**

**Since the last sentences of the introduction section is misleading, we will delete them and discuss it in the discussion section in the revised manuscript.**

2. Methodology of the image processing for identification of green fractions of plots: The authors used the aerial photo data of R, G and B signals for estimating green fraction area of each plot using fairly conventional technique based n trial and error threshold. At least, the authors had to consider to use vegetation indices that have been commonly (almost universally) used, such as NDVI.

**Thank you for your suggestion. Calculation of normalized difference vegetation index (NDVI) requires both red and near-infrared bands. However, the widely used common camera has red, green and blue bands; and the near-infrared band is not available. Therefore, NDVI cannot be used in this study. What's more, Ren et al. (2014) found that the estimation of FVC based on NDVI from a multi-spectral camera is worse than that from common camera, due to the low resolution of multi-spectral camera.**
**We will include this into the discussion section of the revised manuscript.**

3. Individual points:
P3, L13-15: This sentence was unreasonable. Rapid development of lightweight UAVs has been made by multiple demands. Usage for remote sensing is just one of those demands.
**Thank you for your suggestion. We will change the sentence from**
**" UAVs have become a popular platform at a low cost for high precision photography recently."**
**to**
**" UAVs have become a popular platform at a low cost for high precision photography and other applications recently."**

P3, L18-21: If the test in the aim 1) could not provide the favorable results, the authors could not execute the analysis in aim 2). In this case, they could not reach the conclusion described in the title and abstract. Namely, "if so" in between 1) and 2) is unnecessary.

**Thank you for your suggestion. We will delete "if so" in the revised manuscript.**

P4, L9-10 and Figure 2: The method or techniques for obtaining the values of vegeta-tion cover should be explained in the text.
**Thank you for your suggestion. We will add the following sentences in the revised manuscript.**

**"We set up 3 30 m X 30 m plots in each of four types of grassland in 2012, and we set up 9 50 cm X 50 cm quadrats evenly in each plot. We took photo on each quadrat from May 19 to August 30, 2012. The protocol of measurement and estimation of fractional vegetation cover can be found in Section 2.2 and 2.4, respectively."**

P7, L6: "2.6.1" => "2.6.2"
**Thank you for your careful review. we will correct it in the revised manuscript.**

**References:**

**S. Ren, S. Yi, J. Chen, Y. Qin, and X. Wang (2014), Comparisons of alpine grassland fractional vegetation cover estimation using different digital cameras and different image analysis methods, Pratacultural Science, 31, 1007-1013. (In Chinese with English abstract)**

**S. Yi, Z. Zhou, S. Ren, M. Xu, Y. Qin, S. Chen, and B. Ye (2011), Effects of permafrost degradation on alpine grassland in a semi-arid basin on the Qinghai-Tibetan Plateau, Environ. Res. Lett., 6, http://dx.doi.org/10.1088/1748-9326/6/4/045403**

**Z. Zhou , S. Yi, J. Chen, B. Ye, Y. Sheng, G. Wang, and Y. Ding (2015), Responses of alpine grassland to climate warming and permafrost thawing in two basins with different precipitation regimes on the Qinghai-Tibetan Plateau, Arct. Alp. Res., 47, 125-131.**

---

## Author Response (AR1)

Both two reviewers have commented on very important points for improving the contents and messages of your manuscript. Also, they are willing to re-review your revised manuscript. Please revise your manuscript carefully based on the suggestions by reviewers, and make a reply documents to explain how you revised point by point.

We would like to thank you and the reviewers for valuable comments and suggestions. We have made point-by-point responses and revised the manuscript accordingly. The modified text with change track is attached at the end of this response.

**Reviewer 1.**

This article demonstrates the advantage of UAV for monitoring a keystone species "pika" on alpine grassland of Qinghai-Tibetan Plateau. The article is well organized and interesting results are demonstrated, thus this article is suitable for publish in Bio-geosciences. However, particularly in data analyses part in UAV captured images, they seem like too subjective and too classical, in a point of view remote sensing data processing technique. Thus the reviewer feels to need additional data analyses for the improvement of interpretation of the beheivor of "pika" on alpine grassland. Therefore, the reviewer 's recommendation is major revision for this article.

1.Need more objective analyses on UAV's image data processing The authors processed UAV's fine-spatial resolution's data as: identify green fraction areas, (based on green fraction) AGB, SOC, TN estimate by the empirical relation of laboratory analysis. To estimate green fraction areas are based on try & error threshold method by the data of R, G, B signals. The reviewer suggests to vegetation indices (VIs) estimation by bands data used formula, such as NDVI type. As the author's know, most of VIs is consist of combination of red and near-infrared (NIR) signal, but GRVI (green –red)/(green + red) is also one of good indicator to monitor vegetated area's status. For example, if GRVI is higher, target pixel has (maybe) multi-layer structure of grass. By assist of GRVI type VIs, maybe reduce the uncertainty for the conversion from green area fraction (this information only have "one" layer information, i.e., yes or no) to AGB. In addition, supervised and/or unsupervised classifications are also obtained another useful information about the statistical base (e.g., size distribution of patches in which same classified). Maybe two additional data analyses (GRVI type VIs estimation and apply classification for UAV's images) indicate new-insight.

**Reply:**
**Thank you for your suggestion. We used both the green relative vegetation index (GRVI, Figure 1) and excess green index (EGI, Figure 2) to estimate fractional vegetation cover (FVC). Although there are some subtle differences between two methods (see black rectangles in Figure 1 and 2), the overall FVCs are almost the same. Since there are small differences between the FVC values derived from both methods, we do not further establish the relationship between FVC derived from GRVI method and aboveground biomass. In the revised manuscript, we added a description at the end of Section 2.4 and at the beginning of Section 3.1.**

**Section 2.4**

*It is worth mentioning that no true FVCs of grassland at both quadrat and plot scale exist. Results*
*from supervised classification have been usually used as "true" FVC values. For example,*
*supervised classification using WinCAM software was used in Yi et al. (2011), Ren et al. (2014) and*
*Zhou et al. (2015). Yi et al. (2011) found that widely used visual estimation had large variations*
*among different estimators, while WinCAM classification was time-consuming. Ren et al. (2014)*
*found that try & error threshold-based estimation was comparable to that of WinCAM classification*
*and was more efficient than WinCAM. In addition to EGI based threshold method, we also tried*
*green relative vegetation index (GRVI=(G-R)/(G+R)); we did not try the normalized difference*
*vegetation index (NDVI=(NIR-R)/(NIR+R), where NIR is near infrared band), due to lack of NIR*
*band in a common camera.*
**Section 3.1**
*The GFVCs derived using thresholds of EGI and GRVI were similar, with differences less than 1%*
*(Figure not shown here). Therefore, in the following part, we presented results based on EGI*
*threshold.*

[Figure]

Figure 1 Fractional vegetation cover estimation using green relative vegetation index (GRVI)

[Figure]

Figure 2. Fractional vegetation cover estimation using excess green index (EGI)

Another comments

P4 L18 Fujifilim (China) –> Fujifilm (Japan)

**Reply:**

**Thank you for your suggestion, we made the change in the revised manuscript.**

P4 L27 experimental protocol was approved by Department of Qinghai Prataculture –> If can cited document are available (as written in Chinese), please cite in reference.

**Reply:**

**Thank you for your suggestion. The Department of Qinghai Prataculture provided an oral approval of our study, which requires very limited amount of pika-trapping work. Therefore, no formal document is available. We explained it at the end of Section 2.2 in the revised manuscript.**

*The experiment protocol was approved by Department of Qinghai Prataculture (Due to the small size of experiment, only oral approval was granted).*

4.1. Burying ... It is better to change order paragraph stared from P9L26- (Sun et al. (201&) ..) to easy to read.

**Reply:**

**Thank you for your suggestion. We –changed the sentences from**

**"Sun et al. (2016) classified study sites into four classes, i.e. approximately zero pika density (0-15 ha$^{-1}$), low pika density (15-110 ha$^{-1}$), medium pika density (110-200 ha$^{-1}$), and high pika density (200-300 ha$^{-1}$)."**

**to**

**"Sun et al. (2016) classified study sites into four classes: 1) approximately zero pika density**
**(0-15 ha$^{-1}$); 2) low pika density (15-110 ha$^{-1}$); 3) medium pika density (110-200 ha$^{-1}$), and 4)**
**high pika density (200-300 ha$^{-1}$)."**

**Reviewer 2**
Review of "The burying and grazing effects of Plateau pika on alpine grassland are small: A pilot
study in a semi-arid basin on the Qinghai-Tibetan Plateau" by Shuhua Yi and others
In this manuscript, the authors are trying to show the advantages of the UAV remote sensing for
evaluation of plant biomass. Presented data and descriptions include important information. The
aim and concept of the study presented in this manuscript is interesting. However, there are several
weaknesses before considering a publication as an original article in Biogeosciences. I hope
following comments will help the authors to improve the contents and discussions.
1. Relationship between title and main contents of the text: As the authors themselves said in the
final sentence of the introductory section, their aim is not to describe the effects of pika activity on
grassland degradation. But, the title is telling the conclusion of this question. This kind discrepancy
was fatal for the article. In general, "abstract" is describing the essence of the main text, and "title" is
expressing the essence of the abstract, meaning the "conclusion". The aim of this manuscript in
accordance to the line 18-21 is to show the effectiveness of the aerial photos taken by the UAV for
investigation of vegetative conditions. The title should be changed to fit the main point of this
manuscript. Also the parts of the abstract and concluding section should be revised to match the
aim and conclusion.
**Reply:**
**Thank you for your suggestion. There is an important research question, i.e. whether**
**plateau pika causes degradation of grassland, or plateau pika invades after grassland**
**degradation caused by other factors, e.g. climate warming or overgrazing. Meanwhile,**
**changes of grassland include not only changes of biomass and soil carbon and nitrogen,**
**but also changes of species.**
**The current study specifically quantified the burying and grazing effects of pika on**
**grassland at a scale of ~900 m$^2$. This study cannot quantify pika's effects on species and**
**cannot answer the cause-effect relationship between pika and grassland degradation, which**
**requires long-term experiment studies.**
**The sentences at end of Section 1 were used to confine the scope of this study, i.e only to**
**study the direct effects of burying and grazing of pika on biomass, soil carbon and nitrogen.**
**Since they are misleading, we delete them in the revised manuscript.**
2. Methodology of the image processing for identification of green fractions of plots: The authors
used the aerial photo data of R, G and B signals for estimating green fraction area of each plot
using fairly conventional technique based n trial and error threshold. At least, the authors had to
consider to use vegetation indices that have been commonly (almost universally) used, such as
NDVI.
**Reply:**

**Thank you for your suggestion. Calculation of normalized difference vegetation index (NDVI)**
**requires both red and near-infrared bands. However, the widely used common camera has**
**red, green and blue bands; and the near-infrared band is not available. Therefore, NDVI**
**cannot be used in this study.**
**We add the following sentences at the end of Section 2.4.**
*In addition to EGI based threshold method, we also tried green relative vegetation index*
*(GRVI=(G-R)/(G+R)), we did not try the normalized difference vegetation index*
*(NDVI=(NIR-R)/(NIR+R); where NIR is near infrared band), due to lack of NIR band in a common*
*camera.*
3. Individual points:
P3, L13-15: This sentence was unreasonable. Rapid development of lightweight UAVs has been
made by multiple demands. Usage for remote sensing is just one of those demands.
**Reply:**
**Thank you for your suggestion. We changed the sentence from**
**" UAVs have become a popular platform at a low cost for high precision photography**
**recently."**
**to**
**" UAVs have become a popular platform at a low cost for high precision photography and**
**other applications recently."**
P3, L18-21: If the test in the aim 1) could not provide the favorable results, the authors could not
execute the analysis in aim 2). In this case, they could not reach the conclusion described in the title
and abstract. Namely, "if so" in between 1) and 2) is unnecessary.
**Reply:**
**Thank you for your suggestion. We deleted "if so" in the revised manuscript.**
P4, L9-10 and Figure 2: The method or techniques for obtaining the values of vegeta-tion cover
should be explained in the text.
**Reply:**
**Thank you for your suggestion. We added the following sentences at the end of Section 2.1**
**in the revised manuscript.**
*We set up 3 30 m X 30 m plots in each of four types of grassland in 2012, and we set up 9 50 cm X*
*50 cm quadrats evenly in each plot. We took photo on each quadrat from May 19 to August 30,*
*2012 at a height of 1.5 m. The protocol of measurement and estimation of fractional vegetation*
*cover can be found in Section 2.2 and 2.4, respectively.*
P7, L6: "2.6.1" => "2.6.2"
**Reply:**
**Thank you for your careful review. We corrected it in the revised manuscript.**

[revised manuscript text omitted]

a)

d)

e)

f)

g)

b)

NP

OP

BP

VP

m m c)

cm cm h)

m

[Figure]

m m

**Figure 4.**

[Figure]

**Figure 5.**

[Figure]

**Figure 6.**

[Figure]

**Figure 7.**

[Figure]

**Figure 8.**

[Figure]

**Figure 9.**

[Figure]

---

## Author Response (AR2)

Comments to the Author:
Dear Authors,
Your revisions have substantially made with some additional analysis. Those could make the
discussions convincing. One of reviewers suggests one more point to be improved. Please fix it
and finish up your manuscript.
Regards,
Nobuhito Ohte,
Handling Editor
Reviewer #1 suggested as follows:
Please cite below article to describe GRVI.
Motohka, T., Nasahara, K., Oguma, H., and Tsuchida, S. (2010): Utility of Green-Red Vegetation
Index for remote sensing of vegetation phenology. Remote Sensing, 2(10), 2369-2387.
**Reply:**
**We would like to thank suggestions from the editor and reviewers. We added the reference as**
**suggested in the text.**

[revised manuscript text omitted]